# Match-to-Match Variation on High-Intensity Demands in a Portuguese Professional Football Team

**DOI:** 10.3390/jfmk9030120

**Published:** 2024-07-04

**Authors:** José E. Teixeira, Luís Branquinho, Miguel Leal, Ryland Morgans, Andrew Sortwell, Tiago M. Barbosa, António M. Monteiro, Pedro Afonso, Guilherme Machado, Samuel Encarnação, Ricardo Ferraz, Pedro Forte

**Affiliations:** 1Department of Sport Sciences, Polytechnic of Guarda, 6300-559 Guarda, Portugal; 2Department of Sport Sciences, Instituto Politécnico de Bragança, 5300-253 Bragança, Portugal; barbosa@ipb.pt (T.M.B.); mmonteiro@ipb.pt (A.M.M.); samuel01.encarnacao@gmail.com (S.E.); pedromiguel.forte@iscedouro.pt (P.F.); 3SPRINT—Sport Physical Activity and Health Research & Inovation Center, 2001-904 Guarda, Portugal; 4Research Center for Active Living and Wellbeing (Livewell), Instituto Politécnico de Bragança, 5300-252 Bragança, Portugal; 5Research Centre in Sports Sciences, Health Sciences and Human Development, 5001-801 Vila Real, Portugal; sortwellandrew@gmail.com (A.S.); ricardompferraz@gmail.com (R.F.); 6CI-ISCE, Higher Institute of Educational Sciences of the Douro (ISCE Douro), 4560-708 Penafiel, Portugal; luisbranquinho@ipportalegre.pt (L.B.); miguel.leal@iscedouro.pt (M.L.); 7Life Quality Research Center (LQRC-CIEQV), Complexo Andaluz, Apartado 279, 2001-904 Santarém, Portugal; 8Biosciences Scholl of Elvas, Polytechnic Institute of Portalegre, 7300-110 Portalegre, Portugal; 9School of Sport and Health Sciences, Cardiff Metropolitan University, Cardiff CF23 6XD, UK; rmorgans@cardiffmet.ac.uk; 10School of Health Sciences and Physiotherapy, University of Notre Dame Australia, Fremantle, WA 6160, Australia; 11Department of Sports, Exercise and Health Sciences, University of Trás-os-Montes e Alto Douro, 5001-801 Vila Real, Portugal; pmvafonso@gmail.com; 12Department of Athletes’ Integration and Development, Paulista Football Federation (FPF), São Paulo 05614-060, Brazil; machado.guilhermef@gmail.com; 13Department of Pysical Activity and Sport Sciences, Universidad Autónoma de Madrid, Ciudad Universitaria de Cantoblanco, 28049 Madrid, Spain; 14Department of Sports Sciences, University of Beria Interior, 6201-001 Covilhã, Portugal

**Keywords:** physical performance, activity profile, time–motion, match analysis, soccer

## Abstract

The aim of this study was to analyze the match-to-match variation in high-intensity demands from one Portuguese professional football team according to playing positions. Twenty-three male outfield professional football players were observed during eighteen matches of the Portuguese Second League. Time–motion data were collected using Global Positioning System (GPS) technology. Match running performance was analyzed based on the following three playing positions: defenders (DF), midfielders (MF), and forwards (FW). Repeated measures ANOVA was utilized to compare match running performance within each position role, and seasonal running variation. Practical differences were assessed using the smallest worthwhile change (SWC), coefficient of variation (CV), and twice the coefficient of variation (2CV). Significant differences were found among playing positions in total distance covered (*F* = 15.45, *p* < 0.001, η^2^ = 0.33), average speed (*F* = 12.79, *p* < 0.001, η^2^ = 0.29), high-speed running (*F* = 16.93, *p* < 0.001, η^2^ = 0.36), sprinting (*F* = 13.49, *p* < 0.001, η^2^ = 0.31), accelerations (*F* = 4.69, *p* = 0.001, η^2^ = 0.132), and decelerations (*F* = 12.21, *p* < 0.001, η^2^ = 0.284). The match-to-match running performance encompassed TD (6.59%), AvS (8.67%), HSRr (37.83%), SPR (34.82%), ACC (26.92%), and DEC (27.85%). CV values for total distance covered ranged from 4.87–6.82%, with forwards and midfielders exhibiting the greatest and smallest variation, respectively. Midfielders demonstrated the highest match-to-match variation for all other analyzed variables (8.12–69.17%). All playing positions showed significant variation in high-demanding variables (26.94–37.83%). This study presents the initial analysis of match-to-match variation in high-intensity demands within a Portuguese professional football team. Thus, the position’s specificity and context can provide a helpful strategy for evaluating match-to-match running performance, and for recommending individualized training exercises based on the peak and high-intensity demands for each player’s role within the game.

## 1. Introduction

In football, physical performance has been growing over the last four decades at the professional football level [1,2]. This growth has occurred as a result of the dynamic and unpredictable workloads in football, especially during periods of congested match fixtures and the decisive season phases [3]. The monitoring of training and match running performance provides data that can be used by coaches to inform the implementation of strategies to reduce the risk of injury and promote match preparedness [4]. With the advancement of science, monitoring players’ work-rate profiles during training and competition has become more practicable with computer-aided time–motion analysis [5]. This advancement was mainly due to the development of computerized-tracking systems, such as semi-automatized multi-camera systems and position-tracking systems, also known as Global Positioning Systems (GPS) [6,7,8]. Time–motion analysis provided a data collection approach to quantify running activities and energetic expenditure [1,9].

The high-intensity movements are the most important technical–tactical actions, and express the match running performance during the football game. By analyzing match-to-match variation, we can differentiate between game patterns [10,11] and the tactical role [12,13], and therefore discover how they are related to the overall success of the team [14], while being sensitive to training processes and effects [10,15]. Elite players covered 9–14 km in total during an official match, among which the high-intensity activity (speeds > 19.8 km·h^–1^) represents ~8–10% of the total distance completed, since most movement activities were carried out in low-intensity zones [16]. However, there seems to be evidence that the monitoring of only speed-based locomotor variables may not be sufficient to understand match running performance [17]. Accelerations and decelerations frequently occur in a football match-play, requiring greater physiological and neuromuscular demands [18,19]. For instance, the acceleration and sprint profiles of elite football matches have also been characterized [20,21]. Accelerations can contribute to 7–10% of the player workload for all playing positions during a match-play, while decelerations can represent 5–7% [22].

Previous research has also demonstrated an influence of position role in the players’ match demands [23,24,25]. Match-to-match variation has been quantified across numerous national professional leagues, such as the English [26,27,28], Italian [29,30,31], Spanish [32,33], French [34,35,36], German [37,38], Brazilian [39,40], Norwegian [20], Danish [41], and Australian [42,43]. The literature also focused on the European Champions League [14,44,45], the UEFA Cup/Europe League [46,47], and the World Cup [48]. Changes in speed synchronization and physical performance of an elite football were described [10,11]. Additionally, match-to-match variation in speed synchronization increased, particularly during jogging and running, suggesting that accumulated fatigue over the course of the match may impact players’ ability to maintain synchronization at higher speeds [49]. Recently, two studies have added crucial evidence to this research topic in Portuguese professional football contexts [50,51]. Match-related contextual and situational factors have influenced physical demands in Portuguese football leagues, and these effects may differ from other leagues [51]. Therefore, it is important to explore the variation from match to match in Portuguese professional football leagues in order to fit the match-to-match performance profile, especially the high-intensity and peak demands in this context.

In this sense, differences have been reported when analyzing the patterns of high-intensity demands between different competitions [50,51]. It is widely accepted in the literature that an infinity of factors influence these differences, and, therefore, there is an urgent need to obtain more longitudinal investigations that provide new insights on the variability of high-intensity performance between matches in different competitions [23,24,25]. As a consequence of the use of GPS, more detailed analytical assessments can now be carried out, providing an opportunity to assess the match-to-match variability of high-intensity demands in professional football players [52,53]. Although several studies have been carried out, to the best of our knowledge, no studies have been carried out in Portuguese professional competitions, particularly with regard to match running performance. Therefore, the aim of this study was to analyze the match-to-match variation in high-intensity demands from one Portuguese professional football team according to playing positions.

## 2. Materials and Methods

### 2.1. Participants and Match Sample

Twenty-three male professional football players (age: 32.02 ± 1.19 years; height: 1.82 ± 0.01 m; weight: 74.74 ± 0.53 kg) participated in the study. Eighteen Portuguese Second League (Leadman LigaPro^®^, Portuguese Professional Football League, Porto, Portugal) matches (eight home and ten away) during the second phase of 2019–2020 season were studied. Participants were informed of the study design and its requirements, as well as the possible benefits and risks, and gave their consent prior to the start of the study in accordance with the principles of the Declaration of Helsinki for the study of humans. The study was approved by the local ethical committee of the Technical and Scientific Board of the Higher Institute of Educational Sciences of the Douro (ML: 1;11.11.2020).

### 2.2. Experimental Approach to the Problem

The match data correspond to the observations of the seven outfield players on the same team who played each match (n = 128) in a one game per week microcycle. The analysis has only considered the players who were part of the starting lineup and performed the entire match duration. The substituted players and non-starting players were not analyzed. The number of observations per position role were as follows: DF (n = 67), MF (n = 33), and FW (n = 28). The matches (2 × 45′) were performed in official stadiums (FIFA standard; natural grass; ~100 × 70 m), between 10:00 a.m. and 08:00 p.m., and the mean environment temperature was 14.9 ± 5.3 °C.

### 2.3. Data Collection and Procedures

Eligible players for analysis were monitored in each match using a portable GPS throughout the whole match duration (STATSports Apex^®^, Newry, Northern Ireland). The GPS device provides raw position velocity and distance at 10 Hz sampling frequencies, including accelerometer (100 Hz), magnetometer (10 Hz), and gyroscope (100 Hz). Each player wore the micro-technology inside a mini pocket of a custom-made vest supplied by the manufacturer, which was placed on the upper back between both scapulae. All devices were activated 30 min before the match data collection to allow an acceptable clear reception of the satellite signal. Respecting the optimal signal for human movement measurement, the match data considered eight available satellite signals as the minimum for the observations [54]. The validity and reliability of the global navigation satellite systems (GNSS) as the GPS tracking have been well established in the literature [55,56,57]. The current variables and thresholds should consider a small error of around 1–2% reported in the 10 Hz STATSports Apex^®^ units [55].

### 2.4. Measures

The match running performances were obtained using the following time–motion data: TD covered (m), AvS or distance covered per minute (m·min^−1^), HSR distance (m), rHSR distance (m·min^−1^), number of sprints (SPR), number of accelerations (ACC), and number of decelerations (DEC). The APEX Pro Series Software (https://pro.statsports.com/pro-series-software/, accessed on 30 June 2024) provided information about the following locomotor categories above 19.8 km·h^−1^: HSR (19.8–25.1 km·h^−1^) and SPR (>25.1 km·h^−1^). Both acceleration variables (ACC/DEC) considered the movements made in the maximum intensity zone, as follows: ACC (>3 m·s^2^) and DEC (≤3 m·s^2^). The high-intensity activity thresholds were adapted from previous studies [58,59].

### 2.5. Statistical Analysis

Descriptive statistics and the Kolmogorov–Smirnov and Levene’s tests were used to test the normality and homogeneity, wherein a normal distribution was observed. Differences between playing positions were tested with a one-way analysis of variance (ANOVA) for repeated measures. The effect size eta square (η^2^) was computed and interpreted as follows: (i) without effect if 0 < η^2^ ≤ 0.04; (ii) minimum if 0.04 < η^2^ ≤ 0.25; (iii) moderate if 0.25 < η^2^ ≤ 0.64; and (iv) strong if η^2^ > 0.64 [60]. When a significant difference occurred, Bonferroni post hoc tests were used to identify localized effects. Dunnet’s T3 post hoc tests were applied if variances were not homogeneous. Bonferroni post hoc was performed to evaluate TD, rHSR, SPR, and AvS. The Dunnet’s T3 post hoc was executed to ACC and DEC. The coefficient of variation (CV, in %) was quantified to assess the variation across sampled matches [61]. Practical differences were expressed by smallest worthwhile change (SWC) and twice the coefficient of variation (2CV) [62,63]. SWC was calculated by multiplying 0.2 by standard deviation (SD). Then, 2CV was calculated by multiplying 2 × SD. Additionally, the trivial area was calculated from the SWC, determined as 0.2 times the between-playing positions. Statistical significance was set at *p* < 0.05. Data are presented as the mean ± SD. All statistical analyses were conducted using SPSS for Windows Version 22.0 (SPSS Inc., Chicago, IL, USA).

## 3. Results

### 3.1. Match Running Performance

Figure 1 showed the descriptive statistics of the match running performance for each playing position. Playing position significantly impacted all physical load measures under scrutiny, as follows: TD (*F* = 18.33, *p* < 0.001, η^2^ = 0.227), rHSR (*F* = 5.56, *p* = 0.005, η^2^ = 0.082), AvS (*F* = 18.96, *p* < 0.001, η^2^ = 0.233), SPR (*F* = 7.10; p < 0.001, η^2^ = 0.108), ACC (*F* = 7.59, *p* < 0.001, η^2^ = 0.108), and DEC (*F* = 14.06, *p* < 0.001, η^2^ = 0.184). The comparative analysis of data from three groups (DF, MF, and FW) and all participants (ALL) revealed significant differences across various measures. For TD, a statistically significant difference was found (*F* = 18.33, *p* < 0.001, η^2^ = 0.23), with disparities observed between the DF and FW groups (*p* < 0.05), as well as between DF and MF (*p* < 0.001). 

However, no significant difference was noted between MF and FW. Regarding the ratio of rHSR, a significant difference emerged (*F* = 5.56, *p* < 0.05, η^2^ = 0.08), with distinctions between the FW and MF groups (*p* < 0.05), while no significant difference was observed between DF and MF or DF and FW. The AvS exhibited significant differences (*F* = 18.96, *p* < 0.001, η^2^ = 0.23), with disparities noted between DF and FW (*p* < 0.001), as well as between DF and MF (*p* < 0.001), but not between MF and FW. For the number of SPR, a significant difference was evident (*F* = 7.10, *p* < 0.001, η^2^ = 0.10), with distinctions between FW and MF (*p* < 0.001), but not between DF and MF or DF and FW. The count of ACC also displayed significant differences (*F* = 7.59, *p* < 0.001, η^2^ = 0.11), with disparities noted between FW and MF (*p* < 0.001), but not between DF and MF or DF and FW. Finally, for the count of DEC, a significant difference was observed (*F* = 14.06, *p* < 0.001, η^2^ = 0.18), with disparities between DF and FW (*p* < 0.001), as well as between DF and MF (*p* < 0.001), but not between MF and FW.

### 3.2. Match-to-Match Variation on High-Intensity Demands

The variation from match to match and the practical disparities, as indicated by 2CV and SWC, in the metrics of full-game analysis across different game positions are detailed in Table 1. Specifically, the coefficient of variation (CV) values for total distance covered (TD) ranged from 6.52–6.95%, with forwards (FW) exhibiting the most considerable variation, whereas midfielders (MF) demonstrated the least variability. Conversely, for all other variables scrutinized, including average sprint speed (AvS), high-speed running ratio (HSRr), sprint count (SPR), acceleration count (ACC), and deceleration count (DEC), midfielders displayed the highest match-to-match variation, with values of 8.13%, 40.74%, 37.84%, 33.65%, and 30.15%, respectively.

The overall variation observed from match to match encompassed TD (6.59%), AvS (8.67%), HSRr (37.83%), SPR (34.82%), ACC (26.92%), and DEC (27.85%). These findings underscore the nuanced dynamics of match performance across various playing positions, and underscore the importance of considering positional differences in training and tactical strategies. Figure 2 presented the practical differences expressed by coefficient of variation (CV), twice the coefficient of variation (2CV), and smallest worthwhile change (SWC) for match running performance according to positional role.

## 4. Discussion

This study aimed to examine the match-to-match variation in high-intensity demands from one Portuguese professional football team according to playing positions. For TD, AvS, and DEC, as well as for rHSR, SPR, and ACC, positional disparities were shown between DF vs. MF and MF vs. FW. The overall match-to-match variation was as follows: TD (6.59%), AvS (8.67%), HSRr (37.83%), SPR (34.82%), ACC (26.92%), and DEC (27.85%). The research hypothesis, that running performance would differ across playing positions and matches, specifically the high-intensity demands, was confirmed.

Regarding match-to-match variation, the CV ranged from 6.52–6.95%, with forwards exhibiting the highest variability, and midfielders displaying the least [25,64]. Conversely, midfielders demonstrated the highest match-to-match variation across other analyzed variables (i.e., AvS, HSRr, SPR, ACC, and DEC). Differences in high-intensity activity were notably influenced by match outcomes, with forwards exhibiting higher activity levels in winning scenarios, while defenders displayed the opposite trend [23,65]. Additionally, midfielders generally presented higher activity demands compared to defenders and forwards, except for rHSR, ACC, and DEC. Contextual factors, such as match location and home goal percentage, can be controlled in future studies [32,33]. Higher-ranked teams were reported to cover more distance at walking and jogging speeds, with less total distance and high-speed running compared to lower-ranked teams [53,66]. The variations observed in match running across playing positions may be attributed to team and competition ranking, collective behavior, pacing strategies, and contextual factors [67,68]. Competitive level and psychophysiological factors are also likely determinants of match running variability across a season [69,70].

The intensity and distribution of high-intensity efforts during football matches in major European leagues are influenced by various factors, including playing style, tactical approach, and overall competition level [71,72]. Each league exhibits unique characteristics, such as the English Premier League’s fast-paced and physically demanding style, or Serie A’s tactical discipline and defensive organization. However, these observations are generalized, and variations exist within each league and among individual teams [51,71,73]. Moreover, match running performance analysis should consider technical factors, collective behaviors, pre-match cumulative effects, players’ cognitive status, and individual player characteristics [50]. Previous research demonstrated that the TD and high-intensity distance were significantly lower in the second half of the season compared to the first, regardless of the number of matches played, indicating a seasonal decline in these physical performance metrics [74]. However, no significant differences were observed between halves for high-speed running and maximum speed, though these metrics showed variable effect sizes in different matches throughout the season [51,75]. Positional analysis demonstrated that wide midfielders exhibited the greatest variability in total distance, while central defenders showed the highest variability in high-intensity and sprint distances [10,11]. Moreover, offensive performance indicators such as possession and line breaks were negatively correlated with distance covered, suggesting that effective ball control and progression may reduce the physical demands on players [10].

The influence of the league on match running performance in Portuguese teams can vary according to several factors, including the predominant style of play in the league, the level of competition, the tactics adopted by the teams, and even the football culture in Portugal [51,75]. However, it is important to note that the specific characteristics of each league can affect players’ performance in terms of running during matches [76,77]. For example, in a league with a more technical and possession-based style of play, such as the Portuguese Primeira Liga, players may emphasize precision in passing and building up plays, which can influence the amount of high-intensity running during matches. On the other hand, in a league known for being fast-paced and physically intense, players may be more prone to performing sprints and covering greater distances during the game. Additionally, the team’s ranking in the league can also play a role in the variation of running performance [78,79]. Teams struggling at the bottom of the table may be more likely to adopt defensive strategies and run more to defend, while teams at the top of the table may seek a more offensive style of play, which can result in less defensive running and more offensive running [80,81]. In summary, the influence of the league on match running performance in Portuguese teams can be varied and multifaceted, depending on a range of factors, including the league’s style of play, the level of competition, and the team’s position in the standings. Understanding the match-to-match variation in physical performance can provide critical insights to optimize training design [11,13], manage player training load [64,67], and improve overall performance [69,75]. High-intensity variables represent the most decisive movements for performance; however, they are the most critical for players’ health, well-being, and athletic fitness [23,24,25,26]. There are also several practical applications of match-to-match variability for training prescription [71,74]. These data can be used to create personalized training plans for the specific microcirculatory structures and recovery requirements of individual players [73]. They can also be used to monitor match-to-match variation, to distribute physical workload evenly throughout the microcycle, and to determine the peak performance of a particular player for lineup control [21,68]. Other practical applications for strength and conditioning to optimize individual physical qualities [82,83], injury prevention [4,5], and post-match recovery [68] can be applied when using variation between high demands in a football match.

The present study’s limitations include a lack of consideration for different match periods, technical factors, and collective behaviors, as well as uncontrolled pre-match cumulative effects and players’ cognitive status. Additionally, current match data only reflect one team’s data, and cannot be generalized to all Portuguese professional teams. Future research should explore match-to-match running performance across the entire season, different Portuguese football leagues, including women’s football [82,83,84], as well as elite and sub-elite leagues. In addition, integrating technical key indicators [75] and tactical behaviors [52] can provide contextual understanding and inform training and tactical strategies effectively [85]. Furthermore, this study had a lack of internal load measurements (e.g., heart rate profile, perceived exertion, blood lactate) to obtain a more comprehensive picture regarding match-related demands [86]. In fact, exploring the relationship between heart rate variability (HRV), perceived fatigue-recovery status, and susceptibility to musculoskeletal injuries, thereby developing strategies to mitigate injury risks based on physiological markers, is crucial. HRV can be an indicator of players’ stress levels and fatigue, potentially informing decisions on player rotation and rest periods [87,88]. In addition, relating internal load and external load variables allows optimized measurement of intensity, training load, and match running performance [89].

## 5. Conclusions

This study confirmed that playing positions over a season influenced the match-to-match variation. Our match data presented positional differences between DF vs. MF and MF vs. FW for TD, AvS, and DEC, and between DF vs. FW and MF vs. FW for rHSR, SPR, and ACC. This study presents the initial analysis of match-to-match variation in high-intensity demands within a Portuguese professional football team. Thus, the position’s specificity and context can provide a helpful strategy for evaluating match-to-match running performance, and for recommending individualized training exercises based on peak and high-intensity demands for each player’s role within the game. Future research should integrate technical key indicators for performances and tactical behaviors, contextualizing the match running performance with tactical strategy and game model.

## Figures and Tables

**Figure 1 jfmk-09-00120-f001:**
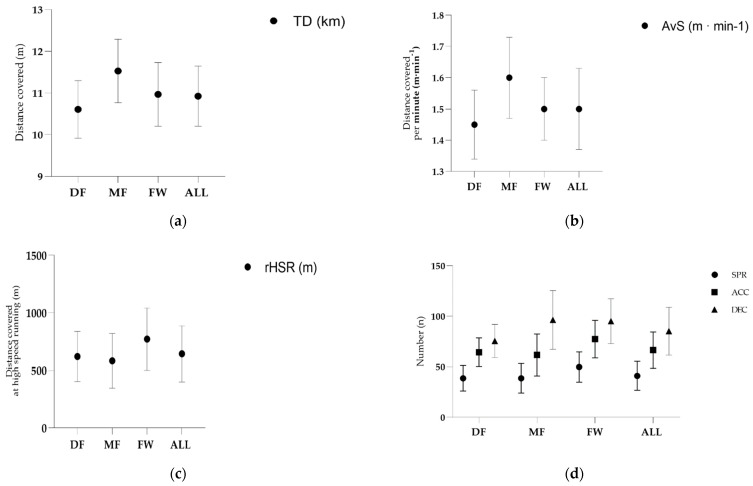
Match running performance according to playing position, as follows: (**a**) total distance covered (km), (**b**) distance covered per minute (m·min^−1^), (**c**) distance covered at relative high-speed running (m), and (**d**) number of sprints, accelerations, and decelerations. Abbreviations: ACC—accelerations; ALL—overall independent position group; AvS—average speed; DEC—Deceleration; DF— defenders; FW—forwards; km—kilometers; m—meters; MF—midfields; m·min^−1^—meters per minute; n—number; rHSR—relative high-speed running; SPR—sprints.

**Figure 2 jfmk-09-00120-f002:**
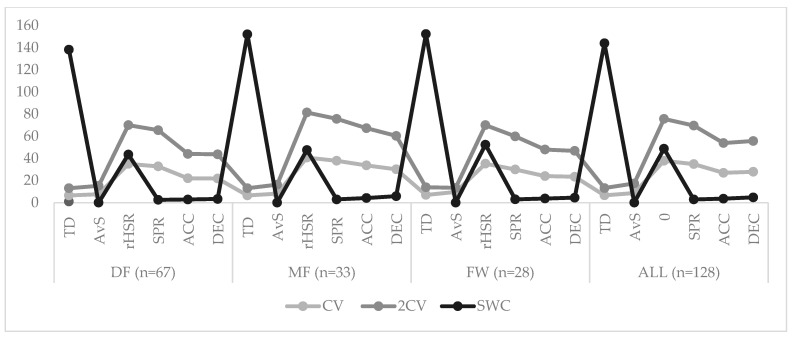
Practical differences expressed by coefficient of variation (CV), twice the coefficient of variation (2CV), and smallest worthwhile change (SWC). Abbreviations: 2CV—twice the coefficient of variation; ACC—accelerations; ALL—overall independent position group; AvS—average speed; CV—coefficient of variation; DEC—decelerations; DF—defenders; FW—forwards; MF—midfielders; rHSR—relative high-speed running; SD—standard deviation; SPR—sprints; SWC—smallest worthwhile change; TD—total distance.

**Table 1 jfmk-09-00120-t001:** Match-to-match variation (CV) and practical differences (2CV, SWC) according to playing position.

Measures	DF (*n* = 67)				MF (*n* = 33)				FW (*n* = 28)				ALL (*n* = 128)			
	CV	SD	2CV	SWC	CV	SD	2CV	SWC	CV	SD	2CV	SWC	CV	SD	2CV	SWC
TD (km)	6.52	691.13	13.04	138.23	6.58	760.36	13.04	152.072	6.95	761.84	13.9	152.37	6.59	720.02	13.18	144.00
AvS (m/min^−1^)	7.59	0.01	15.18	0.002	8.13	0.13	16.26	0.026	6.67	0.1	13.34	0.02	8.67	0.13	17.34	0.03
rHSR (m/min)	35.03	217.29	70.06	43.46	40.74	237.64	81.48	47.528	35.2	261.66	70.4	52.33	37.83	243.6	75.66	48.72
SPR (n)	32.74	12.67	65.48	2.53	37.84	14.65	75.68	2.93	29.93	14.89	59.86	2.98	34.82	14.34	69.64	2.87
ACC (n)	22	14.16	44	2.83	33.65	20.75	67.3	4.15	23.99	18.57	47.98	3.71	26.92	17.91	53.84	3.58
DEC (n)	21.82	16.49	43.64	3.30	30.15	29.07	60.3	5.814	23.4	22.28	46.8	4.46	27.85	23.74	55.7	4.75

Abbreviations: ACC—accelerations; ALL—overall independent position group; AvS—average speed; CV—coefficient of variation; DEC—decelerations; DF—defenders; FW—forwards; MF—midfielders; rHSR—distance covered at high-speed running; SD—standard deviation; SPR—sprints; SWC—smallest worthwhile change; TD—total distance; 2CV—twice the coefficient of variation.

## Data Availability

Data are available under request to the contact author.

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
