# Peer review of "Match-to-Match Variation on High-Intensity Demands in a Portuguese Professional Football Team"

_jfmk, 2024, doi:10.3390/jfmk9030120_

Round 1
Reviewer 1 Report
Comments and Suggestions for Authors
I don't believe that the work is useful for the scientific purpose, the reasoning and postulates, certainly scientifically evaluated and measured, are nothing new in the clinic and preparation of sport and specifically in football. that sporting performance in football was position specific is nothing new, given the biomechanical studies carried out in the last 10 years and new technologies such as GPS or the clinical study of athletes with cardiac variability among all. I emphasize the quality on the applied methodology, the use of ANOVA measures, the time-movement data collected using Global Positioning System (GPS) technology and the statistical study. the purpose of the research, however, is banal and not new
Author Response
Dear Reviewer,
Thank you very much for the time you spent and your feedback on this manuscript. We have made every effort to take on board your recommendations and comments. We hope this revised version and the responses to the comments (kindly refer to our replies below) will meet your requirements. Note that all new changes in the revised manuscript are edited with the Microsoft Word® tracking tool.
Please, check the point-by-point answer in the attachment.
With best regards,
José Eduardo Teixeira.

Reviewer 2 Report
Comments and Suggestions for Authors
I have carefully reviewed the manuscript of José et al. titled: “Match-to-Match Variation on High-Intensity Demands in a Portuguese Professional Soccer Team".
This manuscript aimed to analyze the match-to-match variation in high-intensity demands from the Portuguese professional soccer team according to playing positions. The topic is relevant. However, I have a number of concerns and suggestions that the authors should address prior to a potential publication.
Abstract:
1. Does the entire season of the Portuguese Second League have 18 matches? If not, be cautious about using the term “seasonal variation”. Also, you use both terms “Match-to-Match Variation” and “seasonal variation”, please use one term to keep the consistency.
2. Please add the usefulness of your study for the practical contribution in the last part of the abstract.
Introduction:
1. Line 48-50, “This growth has occurred as a result of the dynamic and unpredictable workloads in soccer, especially during periods of congested match fixtures, which can increase the risk of reduced match preparedness”. The expression is unclear, how does match preparedness increase the risk?
2. Line 59-61, “The high intensity demands can serve as a valid measure of physical performance during the soccer game, as it differentiates certain game patterns, the tactical role, and is therefore related to the overall success of the team, and being sensitive to training processes and effects”. Please rephrase this sentence since it confused the readers.
3. Line 90-92, “Although several studies have been carried out, to the best of our knowledge”. But you mentioned that “until now, no studies have attempted to analyse match-to-match variation running in Portuguese professional soccer leagues” (Line 81-82). It appears controversial, please think about it.
Methods:
1. Line 100, “the study”, Please capitalize the word “the”.
2. Please add the ethical code in this section.
3. Line 109, “in a 1-game week”. Please remove “a” or “1”.
Results:
1. You mentioned “figure 1” first in the result section, please move figure 1 in front of table 1.
2. Also, it would be clear if you can add the “p value” into the figure 1 because you have the note of “p value”, “η2 – eta squared” under the table 1 but you did not show it both in table 1 and figure 1.
3. The abbreviation of “rHSR” in the note under table 1, Please check the spelling because you use “HSRr” rather than “rHSR”.
Discussion:
1. Line 9-14, I suggest the authors remove this sentence and just leave the main result instead of discussion in the first paragraph.
2. Line 22-24, “Contextual factors, such as match location and home-goal percentage, may impact match running performance, although these were not assessed in the present study”. This explanation is not convincing, please reconsider it.
3. Line 40-57, what is your purpose to have a whole paragraph to explain the influence of the leagues on match running performance? It is not really related to your main findings regarding playing positions and seasonal variations in Portuguese Second League.
Comments on the Quality of English Language
There are some points that the authors need to correct the terms and expressions in the text. Please see the comments and suggestions for the Authors.
Author Response

(The authors gave the same response as above.)

Reviewer 3 Report
Comments and Suggestions for Authors
General comment
The Authors evaluated the match-related variations in terms of high intensity actions in élite football players using a GPS system technology. The study covers a popular topic, the rationale is well established, and the procedures are sufficiently detailed. However, there are some points that should be addressed and I hope my comments could be useful in improving manuscript’s quality.
Discussion
Line 60: please consider to add a shorter paragraph regarding the practical applications derived from the Authors’ results that coaches and practitioners could benefit from that.
Line 61: please consider to add some limitations as the lack of internal load measurements as the heart rate profile and the rating of perceived exertion in order to obtain a more comprehensive picture regarding match related-demands.
Here below some references to consider:
- Rago V, Brito J, Figueiredo P, Costa J, Krustrup P, Rebelo A. Internal training load monitoring in professional football: a systematic review of methods using rating of perceived exertion. J Sports Med Phys Fitness. 2020 Jan;60(1):160-171.
- Formenti D, Trecroci A, Cavaggioni L, Caumo A, Alberti G. Heart rate response to a marathon cross-country skiing race: A case study. Sport Sci. Health 2014, 11, 125–128.
Author Response

(The authors gave the same response as above.)

Round 2
Reviewer 3 Report
Comments and Suggestions for Authors
Please consider to insert in the limitation section the lack of internal load measurements (e.g., perceived exertion, blood lactate, heart rate) by adding references to reinforce the statement.
Here below some references to consider:
- Rago V, Brito J, Figueiredo P, Costa J, Krustrup P, Rebelo A. Internal training load monitoring in professional football: a systematic review of methods using rating of perceived exertion. J Sports Med Phys Fitness. 2020 Jan;60(1):160-171.
- Formenti D, Trecroci A, Cavaggioni L, Caumo A, Alberti G. Heart rate response to a marathon cross-country skiing race: A case study. Sport Sci. Health 2014, 11, 125–128.
- McLaren SJ, Macpherson TW, Coutts AJ, Hurst C, Spears IR, Weston M. The Relationships Between Internal and External Measures of Training Load and Intensity in Team Sports: A Meta-Analysis. Sports Med. 2018 Mar;48(3):641-658.
Author Response
Dear Reviewer,
We accepted the reviewer's suggestion and added the recommended references (lines 81–89).
Thank you very much for your comment.

Round 3
Reviewer 3 Report
Comments and Suggestions for Authors
I don't have further comments for Authors.